# Prognostic value of FoxP3 and CTLA-4 expression in patients with oral squamous cell carcinoma

Kazushige Koike[1], Hironari Dehari[1], Kazuhiro Ogi[1], Shota Shimizu[1], Koyo Nishiyama[1], Tomoko Sonoda[2], Takanori Sasaki[1], Takashi Sasaya[1], Kei Tsuchihashi[1], Tadashi Hasegawa[3], Toshihiko Torigoe[4], Hiroyoshi Hiratsuka[1,5], Akihiro Miyazaki[1] *

1 Department of Oral Surgery, Sapporo Medical University School of Medicine, Sapporo, Japan,
2 Department of Public Health, Sapporo Medical University School of Medicine, Sapporo, Japan,
3 Department of Surgical Pathology, Sapporo Medical University School of Medicine, Sapporo, Japan,
4 Department of Pathology, Sapporo Medical University School of Medicine, Sapporo, Japan, 5 Department of Oral and Maxillo-Facial Surgery, Toya Kyokai Hospital, Toyako-cho, Abuta-gun, Japan

* amiyazak@sapmed.ac.jp

## Abstract

### Background

Tumor-infiltrating lymphocytes include tumor-reactive lymphocytes and regulatory T-cells. However, the prognostic value of tumor-infiltrating lymphocytes in oral squamous cell carcinoma (OSCC) remains unclear.

### Methods

We used immunohistochemistry to evaluate the presence of tumor-infiltrating FoxP3+ T-cells and CTLA-4+ cells in four distinct histological compartments (tumor parenchyma and stroma at the tumor center, and parenchyma and stroma at the invasive front) and assessed the association between the prevalence of these cells and the histopathological status of 137 patients with OSCC.

### Results

Five-year overall survival, disease-specific survival, and recurrence-free survival were favorable in patients with high numbers of FoxP3+ T-cells in the parenchyma of the invasive front. Recurrence-free survival and metastasis-free survival were decreased in patients with high numbers of CTLA-4+ cells in the parenchyma of the invasive front.

### Conclusions

The presence of FoxP3+ T-cells in the parenchyma of the invasive front may be a useful prognostic factor. Our results indicate that FoxP3+ T-cells may exert site-specific anti-tumor effects but may not play an immunosuppressive role in OSCC. In addition, our results suggest that CTLA-4+ cells suppress the function of FoxP3+ T-cells and promote anti-tumor immunity in OSCC.

**Data Availability Statement:** All relevant data are within the paper and its Supporting Information files.

**Funding:** The authors received no specific funding for this work.

**Competing interests:** The authors have declared that no competing interests exist.

## Introduction

Oral squamous cell carcinoma (OSCC) is the most common cancer of the oral cavity and represents more than 90% of oral cancers [1]. In most cases, treatment consists of radical surgery combined with radiotherapy and chemotherapy. In advanced OSCC, immunotherapeutic approaches have significantly improved survival [2]. Numerous ongoing preclinical and clinical studies have focused on combinational cancer immunotherapy algorithms that are meant to increase treatment efficiency.

Despite recent progress in diagnosis and therapy, the prognosis for patients with OSCC remains poor. The local tumor microenvironment, which includes tumor cells, extracellular substrates, cytokines, and tumor-infiltrating immune cells, plays essential roles in tumor formation, growth, invasion, and metastasis [3–5]. Patients with higher numbers of inherited defective genes in natural killer (NK) cells were reported to have a higher risk of developing cancer, and inherited defects were associated with tumor immune microenvironment subtypes, recruitment of tumor-infiltrating lymphocytes (TILs), immune checkpoint therapy response, and clinical outcomes [6].

The prognostic significance of TILs is still debated. Many studies on various human tumors have shown a significant association between the presence of TILs and patient survival [7,8]. However, it is important to distinguish between different types of T-lymphocytes because they may play different roles in the tumor microenvironment. For example, TILs include both tumor-reactive lymphocytes and regulatory T-cells (Tregs).

Forkhead box protein P3 (FoxP3) is a transcription factor necessary for induction of immunosuppressive functions in regulatory T-cells. FoxP3 may also be a highly specific marker for Tregs in tumors [9]. Because they can inhibit anti-tumor immunity, high numbers of tumor-infiltrating FoxP3[+] T-cells are associated with poor prognosis. Similar findings have also been reported in OSCC [10]. However, recent studies have challenged this notion, showing that FoxP3[+] T-cells can improve the survival of patients with certain types of tumors [11,12]. Several studies have also shown that increased infiltration of FoxP3[+] T-cells is associated with improved local/regional control and survival in patients with head and neck cancer [13–15]; however, other studies have reported poor survival of patients with malignancies such as non-small cell lung cancer [16].

CTLA-4[+] cells, which express cytotoxic T-lymphocyte-associated protein 4 (CTLA-4) of the immunoglobulin superfamily, bind to B7.1 and B7.2 co-stimulatory molecules [17,18]. The *CTLA-4* gene encodes the CTLA-4 receptor that is transiently expressed on activated T-cells and plays a pivotal role in immune regulation by providing a negative feedback signal to T-cells once an immune response has been elicited. CTLA-4[+] cells participate in Treg-mediated immunosuppression [17,18]. Specific antibodies that block CTLA-4[+] cells have been used to enhance anti-tumor immune responses [19].

Numerous studies have shown that Tregs are recruited into human carcinomas [15,20–22]. However, their roles in anticancer immune responses and how they influence patient prognosis remain unclear. Thus, clarifying the prognostic value of the tumor-infiltrating immune cells will lead to better understanding of the different elements present in the microenvironment of OSCC. Moreover, identifying biomarkers that can be used to specifically predict treatment outcomes is essential for personalized medicine. FoxP3[+] T-cells can inhibit inflammatory processes in the tumor microenvironment, favoring tumor progression. Indeed, tumors of the head and neck are considered inflammatory, and use of a preclinical model has shown that the transfer of Tregs can delay the onset of an inflammation-linked cancer [20,23].

The significance of Treg markers in OSCC is undetermined. The relationship between CTLA-4[+] cells and FoxP3[+] T-cells in OSCC is also unclear. In this study, we examined the

prevalence of CTLA-4[+] cells and FoxP3[+] T-cells in OSCC, and investigated the relationship between these cell types and prognosis. We found that FoxP3[+] T-cells and CTLA-4[+] cells may serve as prognostic factors in OSCC and gained insight into the interplay of these cell types and OSCC.

## Materials and methods

### Patients and tissue samples

This retrospective study was conducted according to the principles stated in the 1964 Declaration of Helsinki and its subsequent versions and was approved by the Institutional Review Board of Sapporo Medical University on September 12, 2017 (No. 292–1116). All study participants provided written informed consent. We used tissue samples collected from patients who were diagnosed with OSCC and who underwent definitive surgery between January 2004 and December 2014 at the Sapporo Medical University Hospital (Table 1). Tissue samples were processed routinely, embedded in paraffin, and sectioned at 4-μm thickness. None of the patients received any form of neoadjuvant chemotherapy or radiotherapy before surgery.

**Table 1. Patient and tumor characteristics.**

| Characteristics | No. of patients | Percentage |
|---|---|---|
| Sex | | |
| Male | 76 | 55.5 |
| Female | 61 | 44.5 |
| Age | | |
| <68 | 63 | 46.0 |
| ≥68 | 74 | 54.0 |
| Anatomical site | | |
| Tongue/Floor of the mouth | 89 | 65.0 |
| Other | 48 | 35.0 |
| Primary tumor | | |
| T1 | 46 | 33.6 |
| T2 | 77 | 56.2 |
| T3/4 | 14 | 10.2 |
| Regional lymph nodes | | |
| N (−) | 108 | 78.8 |
| N (+) | 29 | 21.2 |
| Stage grouping | | |
| Stage I | 42 | 30.7 |
| Stage II | 61 | 44.5 |
| Stage III/IV | 34 | 24.8 |
| Histopathological grading | | |
| Grade 1 | 73 | 53.3 |
| Grade 2 | 59 | 43.1 |
| Grade 3 | 5 | 3.6 |
| Lymphovascular invasion | | |
| Absent | 115 | 83.9 |
| Present | 22 | 16.1 |
| Perineural invasion | | |
| Absent | 125 | 91.2 |
| Present | 12 | 8.8 |

## Immunohistochemistry

The presence of FoxP3[+] T-cells and CTLA-4[+] cells in surgical specimens was evaluated via immunohistochemistry. Briefly, 4-μm serial sections of paraffin-embedded samples were deparaffinized in xylene, soaked in 10 mM citrate buffer (pH 8.0), and autoclaved at 121°C for 10 min for antigen retrieval. Endogenous peroxidase activity was blocked by incubating the sections with 0.3% (v/v) hydrogen peroxide in methanol for 30 min. The sections were then incubated with primary monoclonal antibodies targeting FoxP3[+] T-cells (1:100; clone 236A/E7; Lot: E04270-1631; eBioscience, USA) and CTLA-4[+] cells (1:200; ab227709; Lot: GR3255490-2; Abcam, USA) at 4°C overnight. Secondary antibodies were used as indicated by the EnVision[+] system (Dako REAL™ EnVision™/HRP, Rabbit/Mouse (ENV); Dako, Denmark). Immunolabeling was visualized using diaminobenzidine tetrachloride (Dako REAL™ Substrate Buffer, Dako REAL™ DAB+ Chromogen; Dako, Denmark). The sections were counterstained with hematoxylin, dehydrated, cleared, and mounted (Malinol; MUTO PURE CHEMICALS CO., LTD, Japan). Serial sections were also stained with hematoxylin and eosin (H&E) for morphologic assessment of tumor characteristics. Negative controls were processed in the same manner but were not incubated with the primary antibodies.

## Histopathological and immunohistopathological evaluation

The histological slides were evaluated for lymphovascular invasion, perineural invasion, and histopathological grading. FoxP3[+] T-cells and CTLA-4[+] cells were evaluated using four different areas, including the parenchyma and stroma at the tumor center (TCe), and the parenchyma and stroma at the invasive front (IF). First, FoxP3[+] T-cells and CTLA-4[+] cells were identified under 40× magnification; then, FoxP3[+] T-cells and CTLA-4[+] cells in the four regions of the tumor were counted visually. For counting, we chose the areas with the most intense FoxP3 and CTLA-4 staining density in the four tumor regions and performed counting under 400× magnification. Tumor areas with artifacts and necrotic or apoptotic features were excluded. FoxP3[+] T-cells and CTLA-4[+] cells in the IF were counted in areas containing small clusters or nests at the deepest invading margins. At least three random fields were examined to determine the density of the tumor-infiltrating FoxP3[+] T-cells and CTLA-4[+] cells in each tumor compartment; in cases of heterogeneity, we used a cell count that was most representative of the entire section. Densities of FoxP3[+] T-cells and CTLA-4[+] cells were assessed by three authors (KK, SS, and AM) on a personal computer equipped with DP2-BSW software for an Olympus Microscope with a digital camera (Fig 1). Labels bearing the patient pathological number were covered. We analyzed the prognostic role of the densities of FoxP3[+] T-cells and CTLA-4[+] cells for each of the four different tumor areas. Stratified cell density was also estimated in relation to clinicopathological findings, including sex, age, anatomical sites, primary tumor, regional lymph nodes, and stage grouping. Tumor extent and histopathological grading were classified according to the cancer staging manual of the American Joint Committee on Cancer (AJCC) [24].

We examined the tumor parenchyma and stroma at the tumor center (TCe) and invasive front (IF). FoxP3[+] T-cells and CTLA-4[+] cells were first identified using H&E staining under 40× magnification; then, these cells were counted visually in areas with the greatest densities of FoxP3[+] and CTLA-4[+] immunolabeling under 400× magnification in the four locales of the tumor. Scale bars: 500 μm (center image) and 100 μm (inset images).

## Statistical analyses

Disease-specific survival (DSS) was calculated from the date of definitive surgery to the date of death. Overall survival (OS) was defined as the time between the date of definitive surgery and

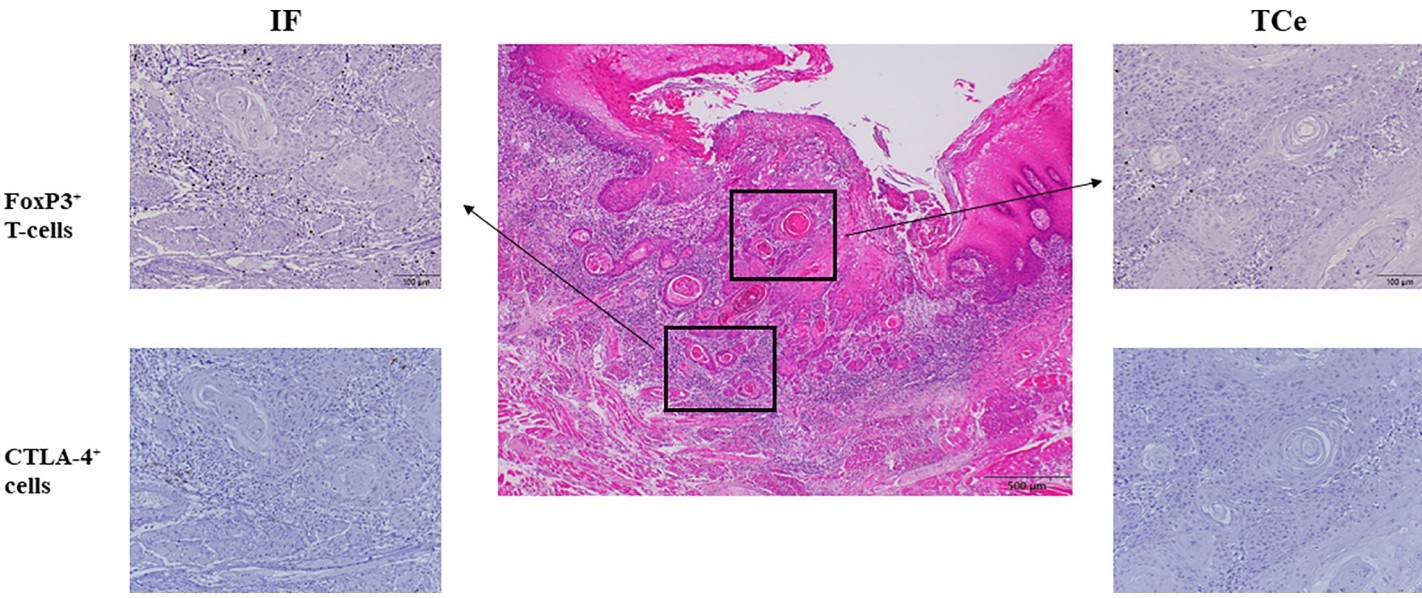

**Fig 1. Densities of FoxP3⁺ T-cells and CTLA-4⁺ cells in the four distinct compartments of OSCC tissues.**

the date of death from any cause. Recurrence-free survival (RFS) was defined as the time between the date of definitive surgery and locoregional or distant tumor recurrence, or death from any cause. Metastasis-free survival (MFS) was defined as the time from first evidence of metastatic disease or death from any cause. In this study, MFS was examined in 108 N0 cases. The median cell density of FoxP3⁺ T-cells or CTLA-4⁺ cells was used as a cutoff point for each compartment and was employed to stratify patients into high or low groups. DSS, OS, RFS, and MFS were analyzed using the Kaplan–Meier method and compared using the log-rank test for each group. Variables that had prognostic potential were evaluated using a stepwise method in Cox regression models. We used the following variables: clinical characteristics (sex, age, anatomical site, and stage grouping), pathological characteristics (histopathological grading, lymphovascular invasion, and perineural invasion), and immunohistochemical findings (infiltrating FoxP3⁺ T-cells and CTLA-4⁺ cells in tumor parenchyma and stroma at the TCe and IF). Two-tailed $P$-values of <0.05 were considered statistically significant. The software Statistical Package for Social Sciences (SPSS), version 23.0, for Windows (IBM, USA) was used for statistical analyses.

## Results

### Patient and tumor characteristics

From January 2004 to December 2014, 221 primary OSCC patients were treated by curative surgery. Of these, 137 patients were treated without neoadjuvant or adjuvant therapy, and sufficient tissue samples from these patients were available for further analyses. The 137 patients with OSCC included 76 men (55.5%) and 61 women (44.5%) with a median age of 68 years (range 33–93 years) at first presentation. Anatomical sites included the tongue/floor of the mouth (89 tumors, 65.0%) and other regions of the oral cavity (48 tumors, 35.0%). Using the AJCC staging classification, patients were classified as follows: 42 patients in stage I (30.7%), 61 in stage II (44.5%), and 34 in stage III/IV (24.8%). Histopathological grading was as follows: 73 patients were grade 1 (53.3%), 59 were grade 2 (43.1%), and five were grade 3 (3.6%). One hundred fifteen patients showed absence (83.9%) and 22 patients showed presence (16.1%) of

lymphovascular invasion; 125 patients showed absence (91.2%) and 12 patients showed presence (8.8%) of perineural invasion. Detailed results are shown in Table 1.

## Relationships between clinicopathological data and survival rates

The median follow-up period for all patients was 79 months (range: 4–164 months). Primary tumors recurred in 16 patients (11.5%), and regional lymph node relapse was found in 25 patients (18.0%). Five-year DSS, OS, and RFS for all patients were 89.6%, 82.4%, and 62.2%, respectively. Detailed results are shown in Table 2. Age and perineural invasion both correlated with OS, DSS, and RFS.

## Density of tumor-infiltrating FoxP3+ T-cells and CTLA-4+ cells

The numbers of tumor-infiltrating FoxP3+ T-cells were as follows: TCe parenchyma (range 0–29; median 2), TCe stroma (range 0–96; median 16), IF parenchyma (range 0–23; median 2), and IF stroma (range 0–122; median 28).

**Table 2. Five-year overall (OS), disease-specific (DSS), and recurrence-free survival (RFS) according to clinicopathological variables.**

| Observed findings | OS | | DSS | | RFS | |
|---|---|---|---|---|---|---|
| | Survival rate (%) | Log-rank test (*P*-value) | Survival rate (%) | Log-rank test (*P*-value) | Survival rate (%) | Log-rank test (*P*-value) |
| Sex | | | | | | |
| Male | 82.9% | 0.93 | 90.8% | 0.89 | 65.8% | 0.34 |
| Female | 83.6% | | 90.2% | | 59.0% | |
| Age | | | | | | |
| <68 | 95.2% | 0.001 | 98.4% | 0.003 | 74.6% | 0.01 |
| ≥68 | 73.0% | | 83.8% | | 52.7% | |
| Anatomical site | | | | | | |
| Tongue/Floor of the mouth | 88.8% | 0.009 | 94.4% | 0.02 | 66.3% | 0.16 |
| Other | 72.9% | | 83.3% | | 56.3% | |
| Primary tumor | | | | | | |
| T1 | 95.7% | 0.001 | 97.8% | 0.03 | 69.6% | 0.12 |
| T2 | 80.5% | | 88.3% | | 62.3% | |
| T3/4 | 57.1% | | 78.6% | | 42.9% | |
| Regional lymph nodes | | | | | | |
| N (–) | 88.0% | 0.001 | 95.4% | <0.001 | 65.7% | 0.17 |
| N (+) | 65.5% | | 72.4% | | 51.7% | |
| Stage grouping | | | | | | |
| Stage I | 97.6% | <0.001 | 100.0% | 0.001 | 71.4% | 0.14 |
| Stage II | 83.6% | | 91.8% | | 63.9% | |
| Stage III/IV | 64.7% | | 76.5% | | 50.0% | |
| Histopathological grading | | | | | | |
| Grade 1 | 87.7% | 0.33 | 94.5% | 0.21 | 71.2% | 0.003 |
| Grade 2 | 78.0% | | 86.4% | | 55.9% | |
| Grade 3 | 80.0% | | 80.0% | | 20.0% | |
| Lymphovascular invasion | | | | | | |
| Absent | 86.1% | 0.05 | 92.2% | 0.14 | 67.0% | 0.01 |
| Present | 68.2% | | 81.8% | | 40.9% | |
| Perineural invasion | | | | | | |
| Absent | 86.4% | <0.001 | 92.8% | 0.001 | 66.4% | 0.001 |
| Present | 50.0% | | 66.7% | | 25.0% | |

**Table 3. Patient distribution according to locations and densities of tumor-infiltrating FoxP3⁺ T-cells.**

| FoxP3⁺ T-cell infiltration locations | | | No. of patients | Percentage |
|---|---|---|---|---|
| TCe | Stroma | Cutoff point = 16 | | |
| | | High: ≥16 | 71 | 51.8 |
| | | Low: <16 | 66 | 48.2 |
| | Parenchyma | Cutoff point = 2 | | |
| | | High: ≥2 | 78 | 56.9 |
| | | Low: <2 | 59 | 43.1 |
| IF | Stroma | Cutoff point = 28 | | |
| | | High: ≥28 | 71 | 51.8 |
| | | Low: <28 | 66 | 48.2 |
| | Parenchyma | Cutoff point = 2 | | |
| | | High: ≥2 | 72 | 52.6 |
| | | Low: <2 | 65 | 47.4 |

IF, invasive front; TCe, tumor center.

The numbers of tumor-infiltrating CTLA-4⁺ cells were as follows: TCe parenchyma (range 0–15; median 1), TCe stroma (range 0–30; median 1), IF parenchyma (range 0–10; median 1), and IF stroma (range 0–24; median 2).

The numbers of infiltrating FoxP3⁺ T-cells and CTLA-4⁺ cells in tumor stroma were greater than those in tumor parenchyma. Detailed results are shown in Tables 3 and 4.

## Relationships between the density of FoxP3⁺ T-cells and survival rates

Patients with OSCC and high density of FoxP3⁺ T-cells in the IF of tumor parenchyma showed significantly increased OS (91.7% vs. 73.8%, p = 0.006), DSS (95.8% vs. 84.6%, p = 0.02), RFS (73.6% vs. 50.8%, p = 0.008), and MFS (87.7% vs. 70.6%, p = 0.02). On the other hand, differences were not statistically significant in the other distinct tumor compartments: TCe parenchyma, TCe stroma, and IF stroma (Figs 2–5).

**Table 4. Patient distribution according to locations and densities of tumor-infiltrating CTLA-4⁺ cells.**

| CTLA-4⁺ cell infiltration locations | | | No. of patients | Percentage |
|---|---|---|---|---|
| TCe | Stroma | Cutoff point = 2 | | |
| | | High: ≥2 | 67 | 48.9 |
| | | Low: <2 | 70 | 51.1 |
| | Parenchyma | Cutoff point = 1 | | |
| | | High: ≥1 | 56 | 40.9 |
| | | Low: <1 | 81 | 59.1 |
| IF | Stroma | Cutoff point = 2 | | |
| | | High: ≥2 | 65 | 47.4 |
| | | Low: <2 | 72 | 52.6 |
| | Parenchyma | Cutoff point = 1 | | |
| | | High: ≥1 | 47 | 34.3 |
| | | Low: <1 | 90 | 65.7 |

IF, invasive front; TCe, tumor center.

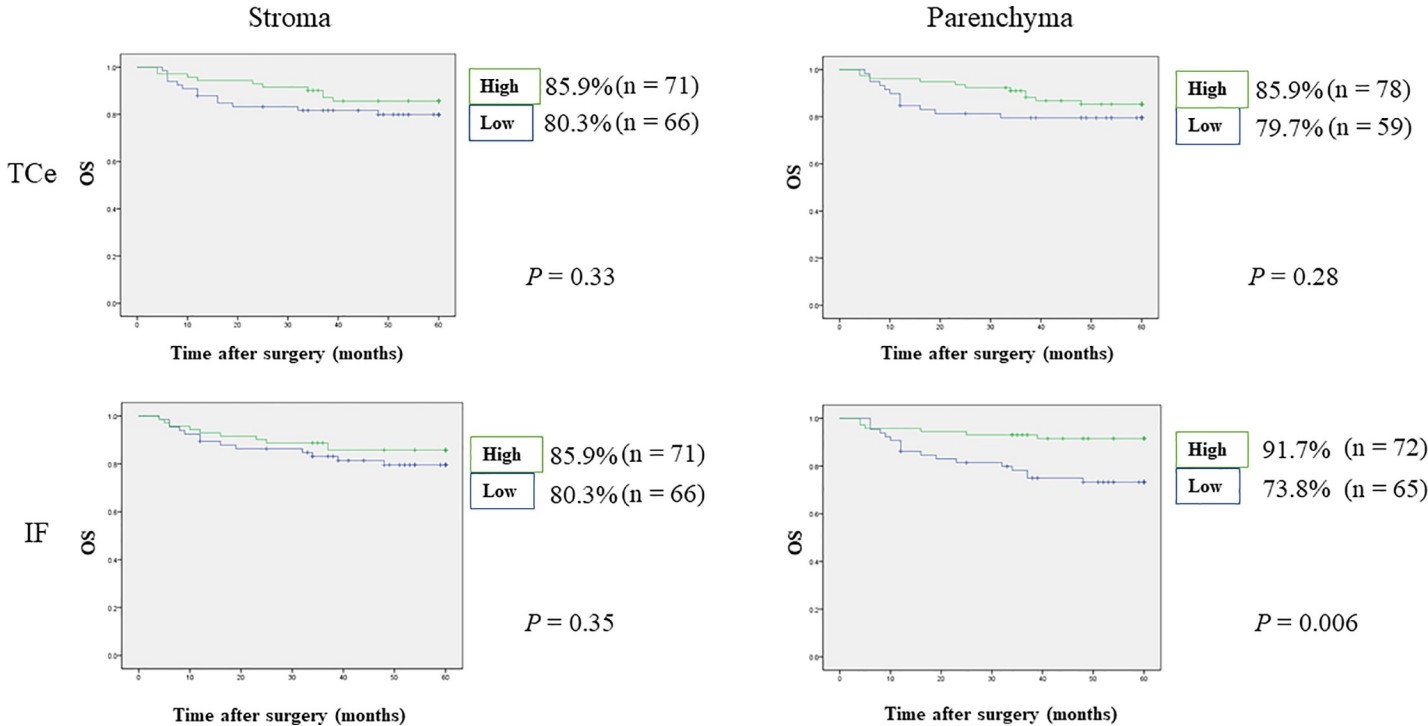

**Fig 2. Relationships between 5-year overall survival and tumor-infiltrating FoxP3⁺ T-cells.** Blue lines indicate the numbers of FoxP3⁺ T-cells below the median, and green lines indicate the numbers of FoxP3⁺ T-cells above the median. Vertical axes indicate the cumulative survival rate, and horizontal axes indicate time in months. Patients with OSCC and high FoxP3⁺ T-cell densities in IF parenchyma showed significantly superior OS compared with patients showing low FoxP3⁺ T-cell densities. DSS, disease-specific survival; IF, invasive front; MFS, metastasis-free survival; OS, overall survival; RFS, relapse-free survival; TCe, tumor center.

Cox multivariate analysis indicated that FoxP3⁺ T-cell density and clinicopathological features could be applied to predict relapse and survival.

The stepwise method in Cox regression models indicated that IF parenchyma was an independent prognostic factor for OS, DSS, RFS, and MFS (OS: HR 0.26, 95% CI 0.10–0.67, p = 0.005, DSS: HR 0.24, 95% CI 0.06–0.89, p = 0.03, RFS: HR 0.48, 95% CI 0.27–0.85, p = 0.01, MFS: HR 0.39, 95% CI 0.15–0.97, p = 0.04). Detailed results are shown in Table 5.

## Relationships between density of CTLA-4⁺ cells and survival rates

Worse DSS was observed in patients with high numbers of CTLA-4⁺ cells in TCe parenchyma (83.9% vs. 95.1%, p = 0.03), but no significant difference in OS or DSS was observed for the other tumor areas. In addition, worse prognoses were observed in patients with high numbers of CTLA-4⁺ cells in IF parenchyma (OS: 78.7% vs. 85.6%, p = 0.31, DSS: 85.1% vs. 93.3%, p = 0.12). RFS was worse in patients with high numbers of CTLA-4⁺ cells in TCe stroma (52.9% vs. 73.1%, p = 0.01) and IF parenchyma (46.8% vs. 71.1%, p = 0.006). Furthermore, MFS was worse in patients with high numbers of CTLA-4⁺ cells in TCe stroma and parenchyma (TCe stroma: 67.9% vs. 90.9%, p = 0.003; TCe parenchyma: 68.6% vs. 84.9%, p = 0.03) and IF parenchyma (63.4% vs. 89.6%, p = 0.001). Detailed results are shown in Figs 6–9. Stepwise methods in Cox regression models indicated that CTLA-4⁺ cells in IF parenchyma were an independent prognostic factor for RFS and MFS (RFS: HR 2.07, 95% CI 1.19–3.60, p = 0.009; MFS: HR 3.55, 95% CI 1.43–8.82, p = 0.006). Detailed results are shown in Table 6.

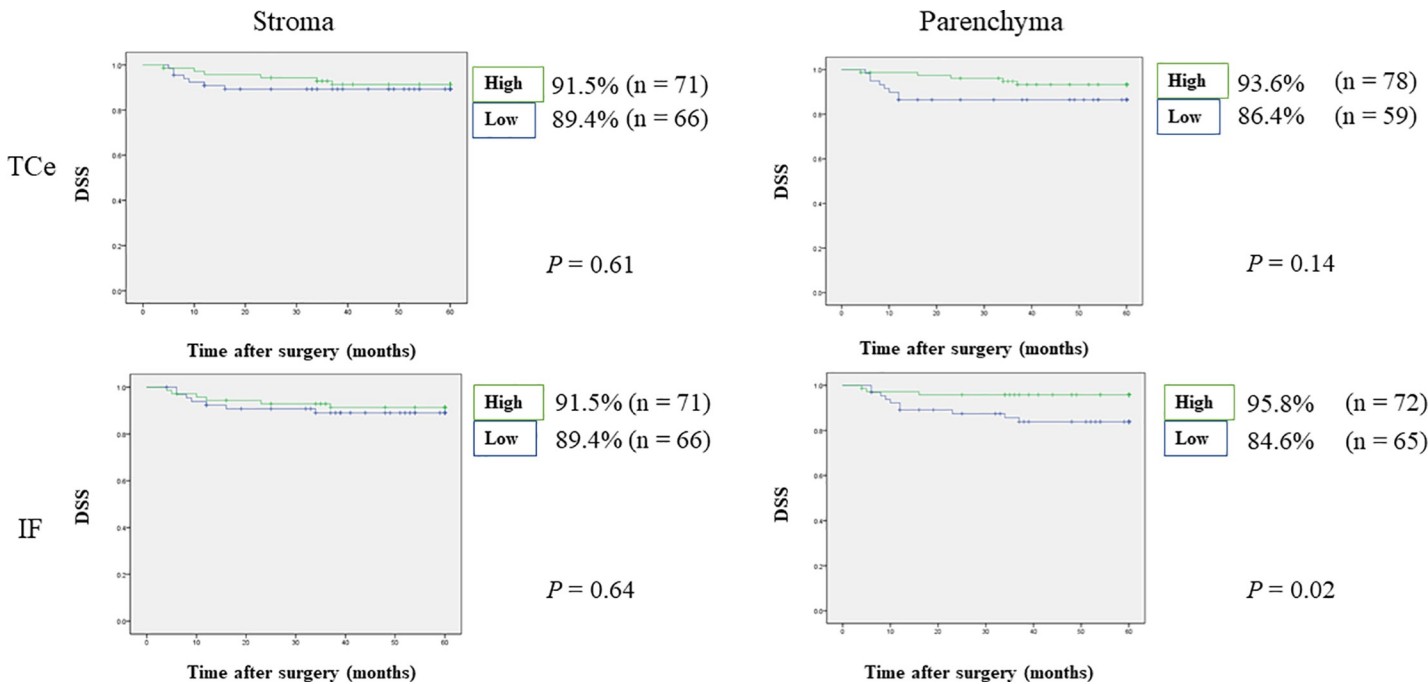

**Fig 3. Relationships between 5-year disease-specific survival and tumor-infiltrating FoxP3+ T-cells.** Patients with OSCC and high FoxP3+ T-cell densities in IF parenchyma showed significantly superior DSS compared with patients showing low FoxP3+ T-cell densities.

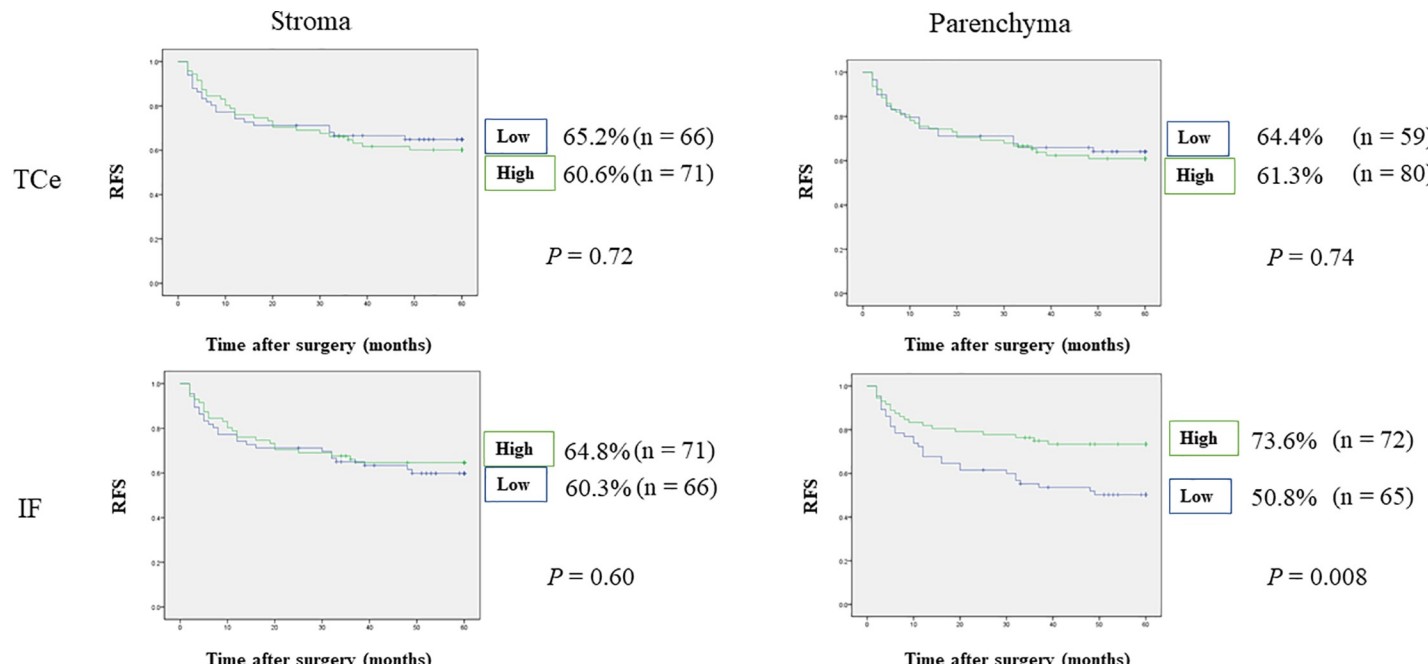

**Fig 4. Relationships between 5-year recurrence-free survival and tumor-infiltrating FoxP3+ T-cells.** Patients with OSCC and high FoxP3+ T-cell densities in IF parenchyma showed significantly superior RFS compared with patients showing low FoxP3+ T-cell densities.

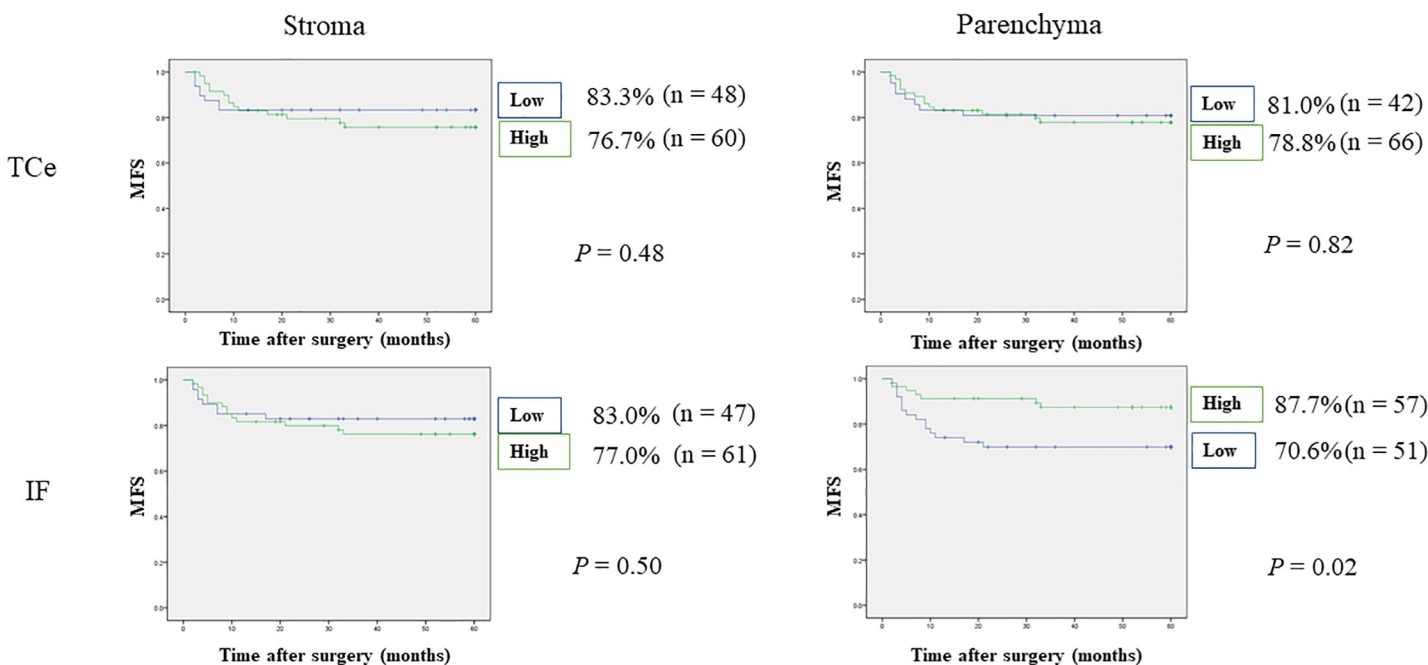

**Fig 5. Relationships between 5-year metastasis-free survival and tumor-infiltrating FoxP3+ T-cells.** Patients with OSCC and high FoxP3+ T-cell densities in IF parenchyma showed significantly superior MFS compared with patients showing low FoxP3+ T-cell densities.

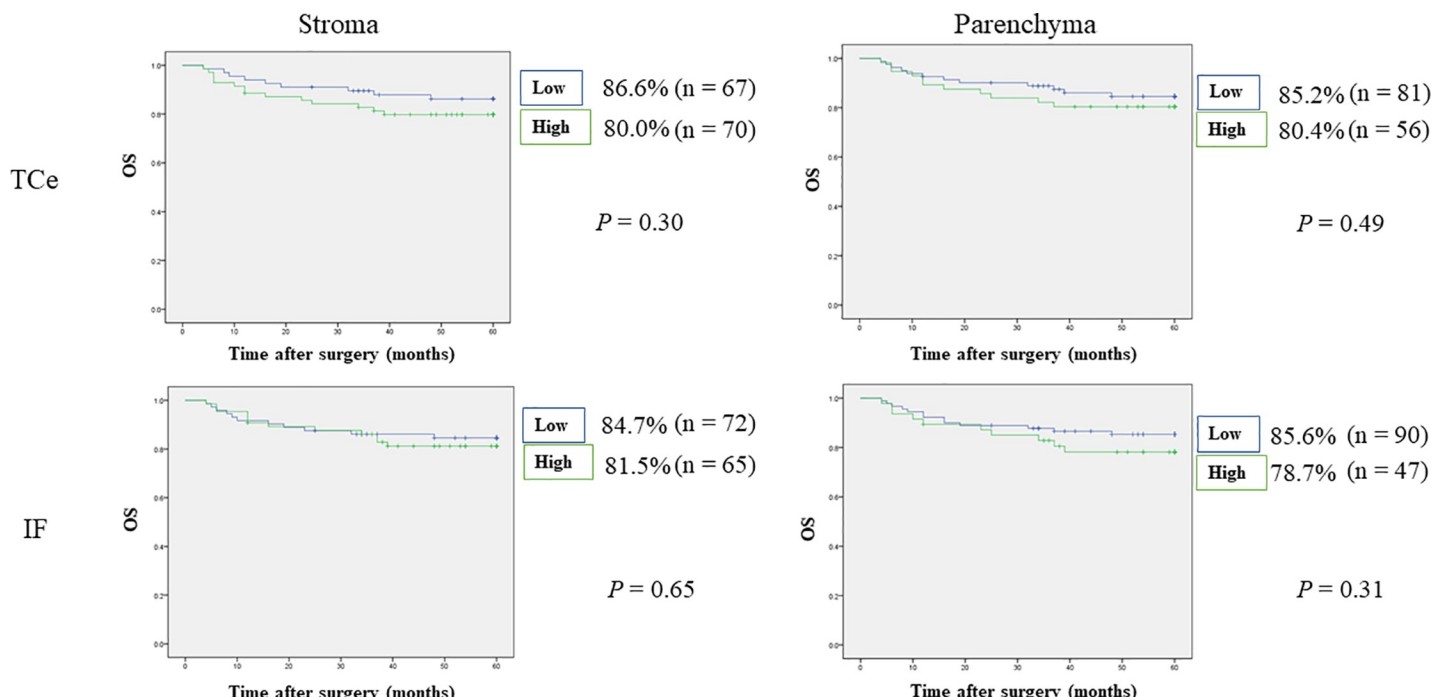

**Fig 6. Relationships between 5-year overall survival and tumor-infiltrating CTLA-4+ cells.** Blue lines represent the numbers of CTLA-4+ cells below the median, and green lines represent the numbers of CTLA-4+ cells above the median. Vertical axes indicate the cumulative survival rate, and horizontal axes indicate time in months. No significant difference in OS was observed. IF, invasive front; OS, overall survival; TCe, tumor center.

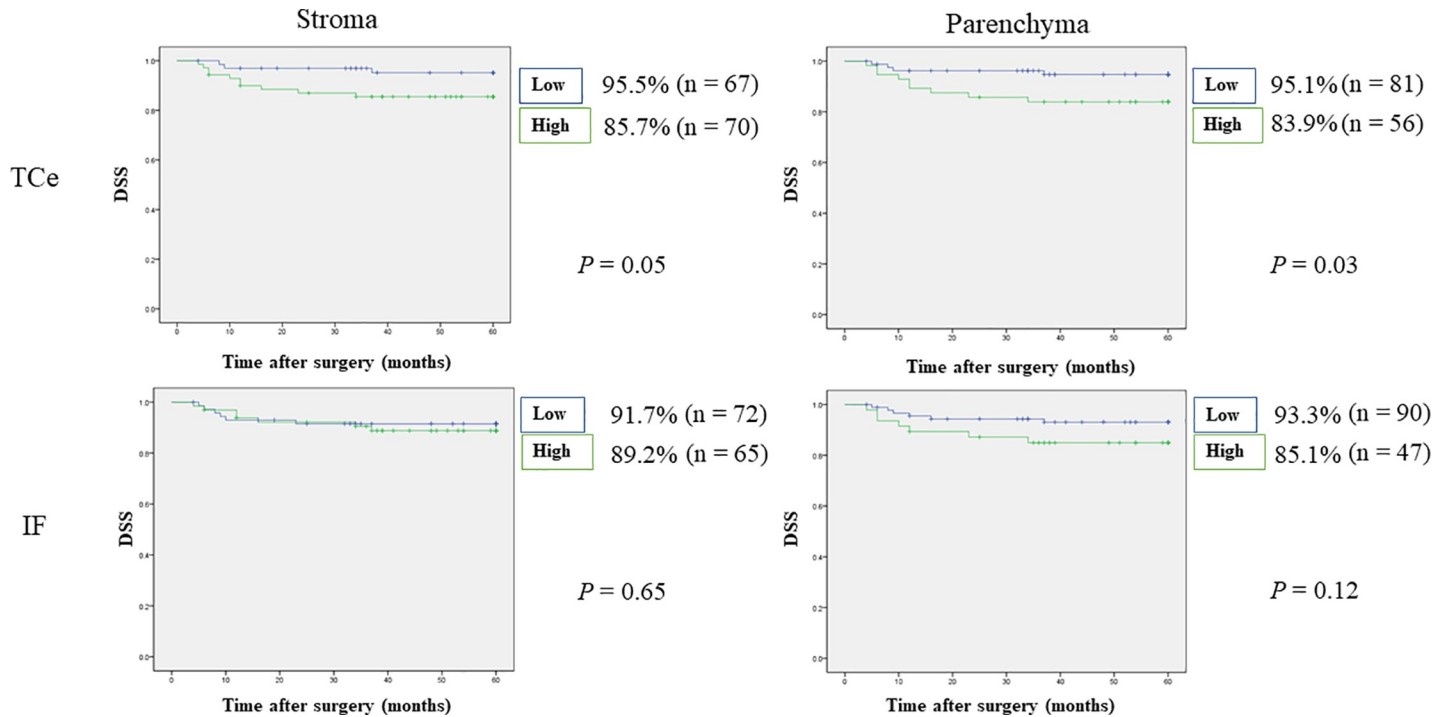

**Fig 7. Relationships between 5-year disease-specific survival and tumor-infiltrating CTLA-4+ cells.** Worse DSS was observed in patients with high numbers of CTLA-4+ cells in TCe parenchyma, but no significant difference was observed for the other tumor areas. DSS, disease-specific survival; IF, invasive front; TCe, tumor center.

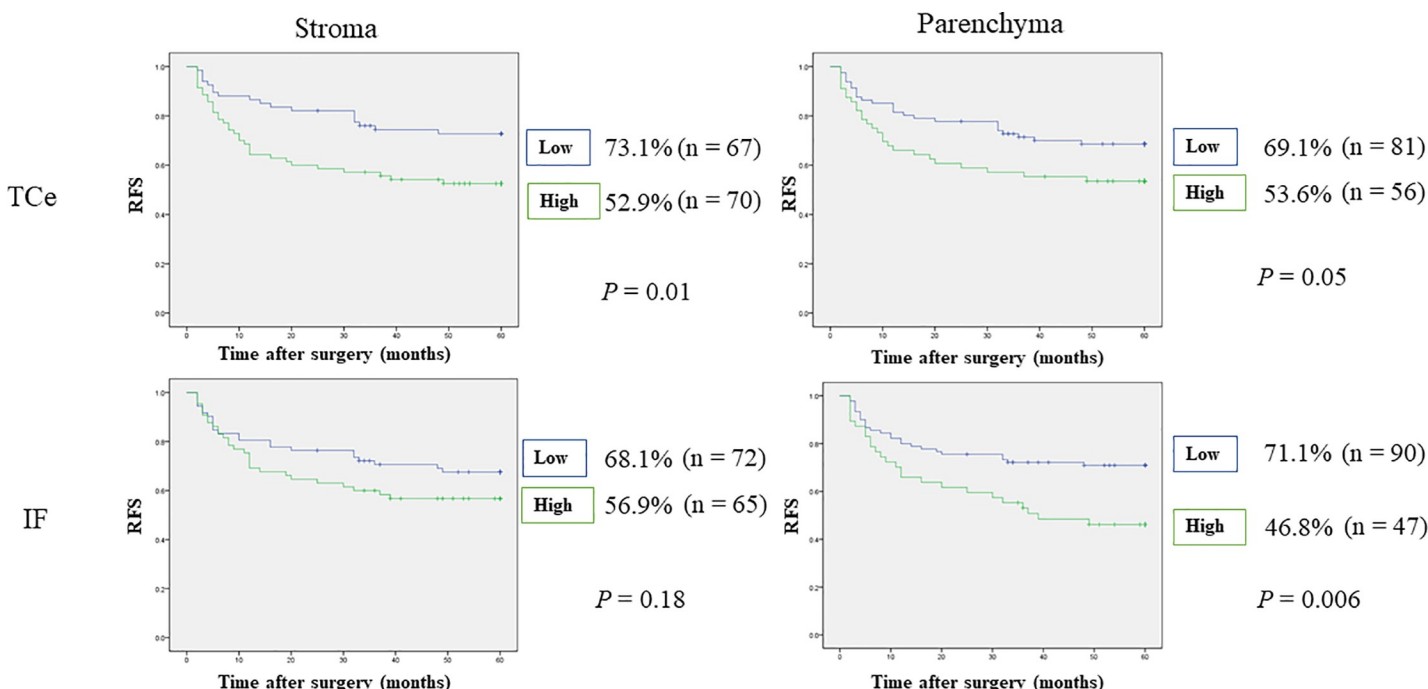

**Fig 8. Relationships between 5-year recurrence-free survival and tumor-infiltrating CTLA-4+ cells.** RFS was decreased in patients with high numbers of CTLA-4+ cells in IF parenchyma or TCe stroma. IF, invasive front; RFS, relapse-free survival; TCe, tumor center.

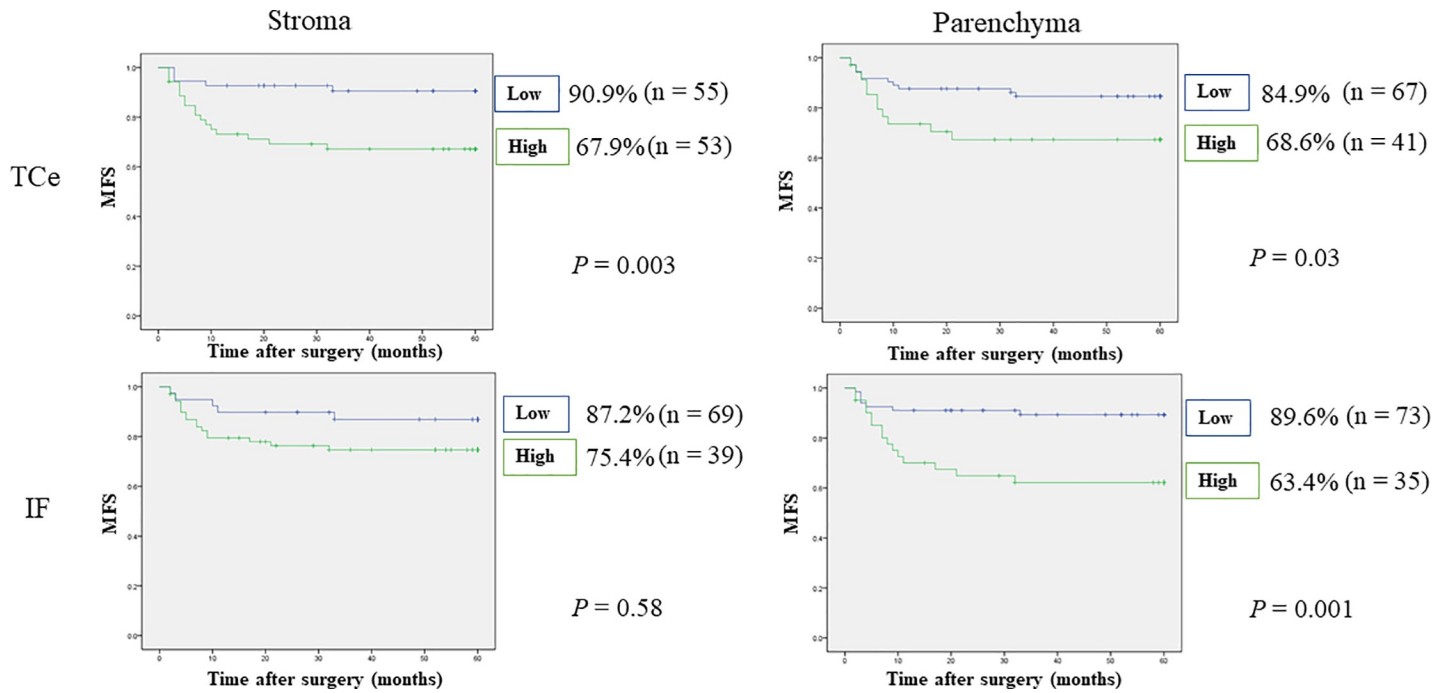

**Fig 9. Relationships between 5-year metastasis-free survival and tumor-infiltrating CTLA-4⁺ cells.** MFS was decreased in patients with high numbers of CTLA-4⁺ cells in TCe parenchyma or stroma and IF parenchyma. IF, invasive front; MFS, metastasis-free survival; TCe, tumor center.

**Table 5. Cox regression models for survival, immunohistochemical, clinical, and pathological findings (tumor-infiltrating FoxP3⁺ T cells).**

| Immunohistochemical, clinical, and pathological findings | | OS | |
| --- | --- | --- | --- |
| | HR | 95% CI | *P*-value |
| IF-Parenchyma | 0.26 | 0.10–0.67 | 0.005 |
| Age | 4.50 | 1.51–13.36 | 0.007 |
| Stage grouping | 2.52 | 1.31–4.82 | 0.005 |
| Perineural invasion | 2.95 | 1.09–7.98 | 0.03 |
| | | DSS | |
| | HR | 95% CI | *P*-value |
| IF-Parenchyma | 0.24 | 0.06–0.89 | 0.03 |
| Age | 10.40 | 1.34–80.83 | 0.02 |
| Stage grouping | 3.29 | 1.26–8.56 | 0.01 |
| | | RFS | |
| | HR | 95% CI | *P*-value |
| IF-Parenchyma | 0.48 | 0.27–0.85 | 0.01 |
| Perineural invasion | 2.54 | 1.17–5.51 | 0.01 |
| | | MFS | |
| | HR | 95% CI | *P*-value |
| IF-Parenchyma | 0.39 | 0.15–0.97 | 0.04 |
| Perineural invasion | 3.79 | 1.05–13.63 | 0.04 |
| Histopathological grading | 2.11 | 1.00–4.44 | 0.04 |

We used the following variables: clinical findings (sex, age, anatomical site, and stage grouping), pathological findings (histopathogical grading, lymphovascular invasion, and perineural invasion), and immunohistochemical findings (infiltrating FoxP3⁺T cells in tumor parenchyma and stroma at TCe and IF). OS, overall survival; DSS, disease-specific survival; RFS, relapse-free survival; MFS, metastasis-free survival; IF, invasive front; TCe, tumor center.

**Table 6. Cox regression models for survival, immunohistochemical, clinical, and pathological findings (tumor-infiltrating CTLA-4⁺ cells).**

| Immunohistochemical, clinical, and pathological findings | | OS | |
|---|---|---|---|
| | HR | 95% CI | *P*-value |
| Age | 4.33 | 1.46–12.86 | 0.008 |
| Stage grouping | 2.48 | 1.29–4.76 | 0.006 |
| Perineural invasion | 2.88 | 1.1–7.51 | 0.03 |
| | | DSS | |
| | HR | 95% CI | *P*-value |
| Age | 9.99 | 1.28–77.47 | 0.028 |
| Stage grouping | 3.64 | 1.44–9.20 | 0.006 |
| | | RFS | |
| | HR | 95% CI | *P*-value |
| IF-Parenchyma | 2.07 | 1.19–3.60 | 0.009 |
| Histopathological grading | 1.86 | 1.15–2.99 | 0.01 |
| | | MFS | |
| | HR | 95% CI | *P*-value |
| IF-Parenchyma | 3.55 | 1.43–8.82 | 0.006 |
| Age | 2.66 | 1.07–6.59 | 0.03 |
| Histopathological grading | 2.41 | 1.26–4.60 | 0.008 |

We used the following variables: clinical findings (sex, age, anatomical site, and stage grouping), pathological findings (histopathological grading, lymphovascular invasion, and perineural invasion), and immunohistochemical findings (infiltrating FoxP3⁺T cells in tumor parenchyma and stroma at TCe and IF). OS, overall survival; DSS, disease-specific survival; RFS, relapse-free survival; MFS, metastasis-free survival; IF, invasive front; TCe, tumor center.

## Discussion

Functions of FoxP3⁺ T-cells are fundamental to the development and maintenance of immune tolerance [25]. FoxP3⁺ T-cells are involved in maintaining immunological tolerance toward host tissues and considered to be suppressors of anti-tumor immune responses [25]. Thus, the presence of FoxP3⁺ T-cells in the tumor microenvironment would indicate an unfavorable prognosis [26,27]. However, the prognostic value of FoxP3⁺ T-cells differs considerably with respect to different types of cancer. High FoxP3⁺ T-cell counts are associated with improved prognosis in patients with head and neck squamous cell carcinoma [13,15]. Our results show that FoxP3⁺ T-cells in IF parenchyma indicated improved prognoses in patients with OSCC. Thus, the abundance of FoxP3⁺ T-cells in IF parenchyma may serve as a useful prognostic factor. Our results also indicate that FoxP3⁺ T-cells may exert site-specific anti-tumor effects in patients with OSCC.

Determining how anti-tumor immunity is regulated in OSCC is critically important. Three mechanisms may be involved in this process. First, FoxP3⁺ T-cells are heterogeneous, involved in both regulatory and non-regulatory functions against tumor immunity [21]. FoxP3⁺ T-cells possibly exert non-regulatory functions in patients with OSCC but identifying them is difficult because uniform immunohistochemical staining can be vague and misleading. Quantitative and qualitative assessment of each FoxP3⁺ T-cell subpopulation, rather than enumeration of the whole FoxP3⁺ T-cell population, may facilitate identifying these different cellular functions. Indeed, the compositions of naïve and effector FoxP3⁺ T-cells and non-regulatory FoxP3⁺ T-cells are disease specific [28]. Targeting a particular FoxP3⁺ T-cell subpopulation, rather than all the FoxP3⁺ T-cells, allows for more effective control of immune responses. For

example, in Treg-mediated cell therapy, in vivo expansion is used to suppress immune responses, while depletion or functional blockade of Tregs is used to enhance immune responses [22].

Second, Tregs may inhibit harmful chronic inflammation. The effectiveness of suppression depends also on T-cell stimulation. Effector T-cells, which are activated by strong co-stimulatory signals or supplemented with growth-promoting cytokines, are refractory to Treg-mediated suppression. Increasing the strength of the T-cell-receptor (TCR) signal received by effector T-cells can also increase their resistance to suppression. These findings suggest that Tregs cannot suppress the production of pro-inflammatory cytokines under conditions that strongly activate effector T-cells. In pro-inflammatory microenvironments, Tregs can be induced to secrete the cytokine IL-17 [29,30]. IL-17[+] FoxP3[+] Tregs, isolated from human peripheral blood, are suppressive in the presence of low TCR stimulation [28,29]; however, IL-17[+] FoxP3[+] Tregs concomitantly lose their immunosuppressive ability and gain the capacity to secrete IL-17 when they are strongly activated in the presence of pro-inflammatory cytokines IL-1β and IL-6 [29]. The loss of suppression, observed in the presence of strong activating factors, is likely due to both increased resistance of effector T-cells and decreased Treg function. This may occur because Tregs are enriched in a T-cell-inflamed environment. This environment, in which functional effector T-cells are attracted by the expression of IFN type I and II, indicates a favorable prognosis [31].

The favorable outcome associated with Tregs may also be due to the opposing effects that depend on the tumor environment; the effect of Tregs is deleterious when they are blocking effector T-cells but beneficial when they are reducing chronic inflammation [32]. In OSCC especially, a tumor is exposed to various microbes and factors such as tobacco and alcohol. In such an environment, Tregs may potentially inhibit harmful chronic inflammation [33]. This notion is supported by findings indicating that stromal localization of Tregs correlates with the patient survival [34]. The results of our present study indicate that in OSCC, the favorable impact of Tregs outweighs their harmful influence.

Third, CTLA-4, which is expressed constitutively on Tregs, critically controls Treg functions and modulates anti-tumor immunity. There are some reports that the suppression of Tregs is regulated by CTLA-4[+] cells [35,36]. Our results show that the presence of CTLA-4[+] cells in IF parenchyma was associated with decreased RFS and MFS. High expression levels of CTLA-4 on Tregs contribute to insufficient Treg depletion, promoting anti-tumor immunity [37]. Our results suggest that CTLA-4[+] cells suppress FoxP3 function in OSCC. CTLA-4[+] cells were found around FoxP3[+] T-cells through double staining (S1 Fig). A recent study in mice has shown that CTLA-4[+] cells are crucial for the suppressive function of FoxP3[+] Tregs in vitro and in vivo [38]. Expression of CTLA-4 on Tregs can promote CD80 and CD86 expression by activated dendritic cells, thereby inhibiting the activation of effector T-cells [39]. Thus, CTLA-4[+] cells with a proliferative gene-expression signature and phenotype are a key feature of OSCC. Targeting CTLA-4[+] cells may lead to new strategies for evoking effective immune responses against OSCC.

Over 100 prognostic biomarkers for OSCC have been investigated and introduced in the past decade. However, none of these biomarkers are presently in clinical use. Common issues with prognostic biomarkers include inadequate validation and paucity of prospective studies. Additionally, multicenter studies are notably sparse, while other researchers have studied small patient cohorts. Our study revealed that FoxP3[+] T-cells and CTLA-4[+] cells may serve as independent prognostic factors in OSCC, but expansion of our knowledge on the functional relationship between FoxP3[+] T-cells and CTLA-4[+] cells is still needed. Future studies should include retrospective and prospective designs with appropriate multivariate analyses of large cohorts.

## Supporting information

**S1 Fig. Double staining of FoxP3+ T-cells and CTLA-4+ cells at invasive front.** Brown and pink staining represent FoxP3+ T-cells and CTLA-4+ cells, respectively. CTLA-4+ cells were localized around FoxP3+ T-cells.
(TIF)

## Acknowledgments

We thank Editage for English language editing of this manuscript.

## Author Contributions

**Conceptualization:** Kazushige Koike, Shota Shimizu, Hiroyoshi Hiratsuka, Akihiro Miyazaki.

**Data curation:** Kazushige Koike, Shota Shimizu.

**Formal analysis:** Kazushige Koike, Hironari Dehari, Kazuhiro Ogi, Tomoko Sonoda.

**Funding acquisition:** Koyo Nishiyama, Akihiro Miyazaki.

**Investigation:** Kazushige Koike, Shota Shimizu.

**Methodology:** Kazushige Koike, Akihiro Miyazaki.

**Project administration:** Hironari Dehari, Kazuhiro Ogi, Akihiro Miyazaki.

**Resources:** Kazushige Koike, Hironari Dehari, Kazuhiro Ogi, Shota Shimizu, Koyo Nishiyama, Takanori Sasaki, Takashi Sasaya, Kei Tsuchihashi, Tadashi Hasegawa.

**Software:** Kazushige Koike, Tomoko Sonoda.

**Supervision:** Tadashi Hasegawa, Toshihiko Torigoe, Akihiro Miyazaki.

**Validation:** Kazushige Koike, Hironari Dehari, Kazuhiro Ogi.

**Visualization:** Kazushige Koike, Shota Shimizu.

**Writing – original draft:** Kazushige Koike, Hironari Dehari, Akihiro Miyazaki.

**Writing – review & editing:** Kazushige Koike, Hironari Dehari, Kazuhiro Ogi, Shota Shimizu, Koyo Nishiyama, Tomoko Sonoda, Takanori Sasaki, Takashi Sasaya, Kei Tsuchihashi, Tadashi Hasegawa, Toshihiko Torigoe, Hiroyoshi Hiratsuka, Akihiro Miyazaki.

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
