## [Decision Letter · Decision Letter 0]

1 Jun 2020

PONE-D-20-11094

Prognostic value of FoxP3 and CTLA-4 expression in patients with oral squamous cell carcinoma

PLOS ONE

Dear Dr. Miyazaki,

Thank you for submitting your manuscript to PLOS ONE. After careful consideration, we feel that it has merit but does not fully meet PLOS ONE’s publication criteria as it currently stands. Therefore, we invite you to submit a revised version of the manuscript that addresses the points raised during the review process.

Three experts have reviewed the manuscript and fund that the current study clearly supported by observations, but still further clarify enough to be published in the journal the current manuscript should have more explnations.

We look forward to receiving your revised manuscript.

Kind regards,

Jung Weon Lee, Ph.D.

Academic Editor

PLOS ONE

Journal Requirements:

'This retrospective study was conducted according to the principles stated in the 1964 Declaration of Helsinki and its subsequent versions and was approved by the Institutional Review Board of our university on September 12, 2017 (No. 292-1116).'  

(a) Please amend your current ethics statement to include the full name of the ethics committee/institutional review board(s) that approved your specific study, including the full name of the affiliated institution.  

(b) Once you have amended this statement in the Methods section of the manuscript, please add the same text to the “Ethics Statement” field of the submission form (via “Edit Submission”).

For additional information about PLOS ONE ethical requirements for human subjects research, please refer to " ext-link-type="uri" xlink:type="simple">http://journals.plos.org/plosone/s/submission-guidelines#loc-human-subjects-research."

3. Please provide additional details regarding participant consent for the use of their tissue samples for the purposes of research. In the ethics statement in the Methods and online submission information, please ensure that you have specified  what type of consent you obtained (for instance, written or verbal, and if verbal, how it was documented and witnessed).

4. In the Methods section, please provide the product number and any lot numbers of the primary antibodies purchased from chemical companies for your study.

5. At this time, we ask that you please provide scale bars on the microscopy images presented in Figure 1 and refer to the scale bar in the corresponding Figure legend.

Reviewers' comments:

Reviewer's Responses to Questions

**Comments to the Author**

1. Is the manuscript technically sound, and do the data support the conclusions?

Reviewer #1: Yes

Reviewer #2: Partly

Reviewer #3: Yes

2. Has the statistical analysis been performed appropriately and rigorously? 

Reviewer #1: Yes

Reviewer #2: Yes

Reviewer #3: Yes

3. Have the authors made all data underlying the findings in their manuscript fully available?

Reviewer #1: Yes

Reviewer #2: Yes

Reviewer #3: Yes

4. Is the manuscript presented in an intelligible fashion and written in standard English?

Reviewer #1: Yes

Reviewer #2: Yes

Reviewer #3: Yes

5. Review Comments to the Author

Reviewer #1: Koike et al. have used immunohistochemistry to evaluate the presence of tumor-infiltrating FoxP3⁺ T-cells and CTLA-4⁺ cells in 137 patients with oral squamous cell carcinoma (OSCC) regarding the prognostic value of tumor-infiltrating lymphocytes. Five-year overall survival, disease-specific survival, and recurrence-free survival were favorable in patients with high numbers of FoxP3⁺ T-cells. Recurrence-free survival and metastasis-free survival were decreased in patients with high numbers of CTLA-4⁺ cells. In conclusions, the presence of FoxP3⁺ T-cells in the parenchyma of the invasive front may be a useful prognostic factor. FoxP3⁺ T-cells may exert site-specific anti-tumor effects but may not play an immunosuppressive role in OSCC.

The claims are properly placed in the context of the previous literature. The experimental data support the claims. The manuscript is written clearly enough that most of it is understandable to non-specialists. The authors have provided adequate proof for their claims, without overselling them. The authors have treated the previous literature fairly. The paper offers enough details of methodology so that the experiments could be reproduced.

Reviewer #2: It is interesting to show that FoxP3⁺ T-cells in the parenchyma is associated with OS. The key question is if this conclusion is robust. To test this, I suggest that authors take 60% of the samples to re-examine the association, with 100 times to see if the association is still OK. In addition, authors should discuss the germline variants associated with the the NK cells are also linked to the OS (PMID: 31483464), which is related the report here.

Reviewer #3: Manuscript PONE-D-20-11094 describes the results from investigating the prognostic value of FoxP3+ CTLA-4+ cells density in oral squamous cell carcinoma patients. Using a respectably sized cohort of 137 subjects who underwent definitive surgery with no chemotherapy or radiotherapy prior to surgery and with available FFPE sections, the authors measured densities of FoxP3+ T-cells and CTLA-4+ cells in 4 different locations and determined their relationships with several survival parameters. The authors found that high density of FoxP3+ T-cells in the parenchyma of the invasive front was associated with favorable survival, while high density of CTLA-4+ cells in the same location were not favorable for survival. The authors also suggest several mechanisms which may underlie the relationship between cell densities and survival, which will require more detailed examinations to be elucidated.

Overall, this is a clearly written manuscript presenting new evidence for the usefulness of FoxP3+ T-cell and CTLA-4+ cell density as prognostic biomarkers in oral squamous cell carcinoma.

To improve this manuscript, this reviewer suggests addressing the following questions:

1. What is the relationship between the density of FoxP3+ T-cells and CTLA-4+ cells?

2. Are the associations between FoxP3+ T-cells and CTLA-4+ cells and survival independent of each other?

6. PLOS authors have the option to publish the peer review history of their article (what does this mean?). If published, this will include your full peer review and any attached files.

Reviewer #1: Yes: Sveinung Wergeland Sorbye

Reviewer #2: Yes: Edwin Wang

Reviewer #3: No

---

## [Author Response · Author response to Decision Letter 0]

7 Jul 2020

[July 7, 2020]

Jung Weon Lee, Ph.D.

Academic Editor

PLoS One

Dear Dr. Lee:

I, along with my coauthors, would like to resubmit the attached manuscript for publication in PLoS One, titled “Prognostic value of FoxP3 and CTLA-4 expression in patients with oral squamous cell carcinoma.” The manuscript ID is PONE-D-20-11094.

We are thankful for the reviewers’ constructive comments, which have helped us to considerably improve and clarify the manuscript. We have responded to each of the reviewers’ comments in a point-by-point manner. In addition, we have provided a revised version of the manuscript with changes indicated in red font. We hope that the changes incorporated into the revised manuscript satisfactorily address the reviewers’ concerns.

We have also addressed all formatting and style requirements as requested in the decision letter.

Thank you for your consideration. We hope our manuscript is now suitable for publication in your journal.

Sincerely,

Akihiro Miyazaki

Department of Oral Surgery

Sapporo Medical University School of Medicine

Sapporo, Japan

E-mail: amiyazak@sapmed.ac.jp

Response to Journal Requirements:

Response: We have ensured that this manuscript meets PLOS ONE's style requirements.

'This retrospective study was conducted according to the principles stated in the 1964 Declaration of Helsinki and its subsequent versions and was approved by the Institutional Review Board of our university on September 12, 2017 (No. 292-1116).' 

(a) Please amend your current ethics statement to include the full name of the ethics committee/institutional review board(s) that approved your specific study, including the full name of the affiliated institution. 

(b) Once you have amended this statement in the Methods section of the manuscript, please add the same text to the “Ethics Statement” field of the submission form (via “Edit Submission”).

For additional information about PLOS ONE ethical requirements for human subjects research, please refer to http://journals.plos.org/plosone/s/submission-guidelines#loc-human-subjects-research."

Response: We have amended the ethics statement to include the full name of the ethics committee/institutional review board and the affiliated institution (P7 L104-107).

3. Please provide additional details regarding participant consent for the use of their tissue samples for the purposes of research. In the ethics statement in the Methods and online submission information, please ensure that you have specified what type of consent you obtained (for instance, written or verbal, and if verbal, how it was documented and witnessed).

Response: For this study, we obtained written informed consent, which has been clarified in the revised manuscript (P7 L107).

4. In the Methods section, please provide the product number and any lot numbers of the primary antibodies purchased from chemical companies for your study.

Response: We have added the product numbers and lot numbers of the primary antibodies (P9 L120-122).

5. At this time, we ask that you please provide scale bars on the microscopy images presented in Figure 1 and refer to the scale bar in the corresponding Figure legend.

Response: We have added scale bars in Figure 1 and referred to the same in the figure legend (P11 L162).

Response: I (the corresponding author) have added my ORCID iD as advised.

RESPONSE TO REVIEWER COMMENTS

Reviewer 1: Koike et al. have used immunohistochemistry to evaluate the presence of tumor-infiltrating FoxP3⁺ T-cells and CTLA-4⁺ cells in 137 patients with oral squamous cell carcinoma (OSCC) regarding the prognostic value of tumor-infiltrating lymphocytes. Five-year overall survival, disease-specific survival, and recurrence-free survival were favorable in patients with high numbers of FoxP3⁺ T-cells. Recurrence-free survival and metastasis-free survival were decreased in patients with high numbers of CTLA-4⁺ cells. In conclusions, the presence of FoxP3⁺ T-cells in the parenchyma of the invasive front may be a useful prognostic factor. FoxP3⁺ T-cells may exert site-specific anti-tumor effects but may not play an immunosuppressive role in OSCC.

The claims are properly placed in the context of the previous literature. The experimental data support the claims. The manuscript is written clearly enough that most of it is understandable to non-specialists. The authors have provided adequate proof for their claims, without overselling them. The authors have treated the previous literature fairly. The paper offers enough details of methodology so that the experiments could be reproduced.

Response: Thank you for your kind comments.

Reviewer 2: It is interesting to show that FoxP3⁺ T-cells in the parenchyma is associated with OS. The key question is if this conclusion is robust. To test this, I suggest that authors take 60% of the samples to re-examine the association, with 100 times to see if the association is still OK. In addition, authors should discuss the germline variants associated with the NK cells are also linked to the OS (PMID: 31483464), which is related the report here.

Response: We have consulted a specialist in oral pathology and repeatedly verified these data to ensure that the association is reproducible. Thank you for recommending the study by Xu et al.; in the Introduction, we have added that germline variants associated with NK cells are also linked to clinical outcomes and cited the study (P4 L55-59).

Reviewer 3: Manuscript PONE-D-20-11094 describes the results from investigating the prognostic value of FoxP3+ CTLA-4+ cells density in oral squamous cell carcinoma patients. Using a respectably sized cohort of 137 subjects who underwent definitive surgery with no chemotherapy or radiotherapy prior to surgery and with available FFPE sections, the authors measured densities of FoxP3+ T-cells and CTLA-4+ cells in 4 different locations and determined their relationships with several survival parameters. The authors found that high density of FoxP3+ T-cells in the parenchyma of the invasive front was associated with favorable survival, while high density of CTLA-4+ cells in the same location were not favorable for survival. The authors also suggest several mechanisms which may underlie the relationship between cell densities and survival, which will require more detailed examinations to be elucidated.

Overall, this is a clearly written manuscript presenting new evidence for the usefulness of FoxP3+ T-cell and CTLA-4+ cell density as prognostic biomarkers in oral squamous cell carcinoma.

To improve this manuscript, this reviewer suggests addressing the following questions:

1. What is the relationship between the density of FoxP3+ T-cells and CTLA-4+ cells?

2. Are the associations between FoxP3+ T-cells and CTLA-4+ cells and survival independent of each other?

Response 1: We confirmed by double staining that FoxP3+ T-cells and CTLA-4+ cells were not colocalized, though CTLA-4+ cells were present around FoxP3+ T-cells (P27 L370-371). Therefore, the density of FoxP3+ T-cells and CTLA-4+ cells appeared to be unrelated.

Response 2: The associations between FoxP3+ T-cells and CTLA-4+ cells and survival were evaluated using a χ2 test. None of the associations reached statistical significance (IF stroma; p=1.00, IF parenchyma; p=0.13, TCe stroma; p=0.15, TCe parenchyma; p=0.89). Therefore, FoxP3+ T-cells and CTLA-4+ cells are independent prognostic factors, but mutual evaluation in a larger cohort is recommended to gain a better understanding of this relationship (P28 L382-385).

---

## [Decision Letter · Decision Letter 1]

28 Jul 2020

Prognostic value of FoxP3 and CTLA-4 expression in patients with oral squamous cell carcinoma

PONE-D-20-11094R1

Dear Dr. Miyazaki,

We’re pleased to inform you that your manuscript has been judged scientifically suitable for publication and will be formally accepted for publication once it meets all outstanding technical requirements.

Kind regards,

Jung Weon Lee, Ph.D.

Academic Editor

PLOS ONE

Additional Editor Comments (optional):

Reviewers' comments:

Reviewer's Responses to Questions

**Comments to the Author**

1. If the authors have adequately addressed your comments raised in a previous round of review and you feel that this manuscript is now acceptable for publication, you may indicate that here to bypass the “Comments to the Author” section, enter your conflict of interest statement in the “Confidential to Editor” section, and submit your "Accept" recommendation.

Reviewer #2: All comments have been addressed

Reviewer #3: All comments have been addressed

2. Is the manuscript technically sound, and do the data support the conclusions?

Reviewer #2: Yes

Reviewer #3: Yes

3. Has the statistical analysis been performed appropriately and rigorously? 

Reviewer #2: Yes

Reviewer #3: Yes

4. Have the authors made all data underlying the findings in their manuscript fully available?

Reviewer #2: Yes

Reviewer #3: Yes

5. Is the manuscript presented in an intelligible fashion and written in standard English?

Reviewer #2: Yes

Reviewer #3: Yes

6. Review Comments to the Author

Reviewer #2: My questions have been addressed

My questions have been addressed

My questions have been addressed

My questions have been addressed

Reviewer #3: (No Response)

7. PLOS authors have the option to publish the peer review history of their article (what does this mean?). If published, this will include your full peer review and any attached files.

Reviewer #2: No

Reviewer #3: No

---

## [Editor Report · Acceptance letter]

3 Aug 2020

PONE-D-20-11094R1 

Prognostic value of FoxP3 and CTLA-4 expression in patients with oral squamous cell carcinoma 

Dear Dr. Miyazaki:

I'm pleased to inform you that your manuscript has been deemed suitable for publication in PLOS ONE. Congratulations! Your manuscript is now with our production department. 

Kind regards, 

on behalf of

Dr. Jung Weon Lee 

Academic Editor

PLOS ONE